# The Veterinary Anti-Parasitic Selamectin Is a Novel Inhibitor of the *Mycobacterium tuberculosis* DprE1 Enzyme

**DOI:** 10.3390/ijms23020771

**Published:** 2022-01-11

**Authors:** José Manuel Ezquerra-Aznárez, Giulia Degiacomi, Henrich Gašparovič, Giovanni Stelitano, Josè Camilla Sammartino, Jana Korduláková, Paolo Governa, Fabrizio Manetti, Maria Rosalia Pasca, Laurent Roberto Chiarelli, Santiago Ramón-García

**Affiliations:** 1Department of Microbiology, Faculty of Medicine, University of Zaragoza, 50009 Zaragoza, Spain; jmezquerraa@unizar.es; 2Department of Biology and Biotechnology “Lazzaro Spallanzani”, University of Pavia, 27100 Pavia, Italy; giulia.degiacomi@unipv.it (G.D.); giovanni.stelitano01@universitadipavia.it (G.S.); jose.sammartino@iusspavia.it (J.C.S.); mariarosalia.pasca@unipv.it (M.R.P.); 3Department of Biochemistry, Faculty of Natural Sciences, Comenius University in Bratislava, 842 15 Bratislava, Slovakia; gasparovic2@uniba.sk (H.G.); jana.kordulakova@uniba.sk (J.K.); 4Department of Biotechnology, Chemistry and Pharmacy, University of Siena, 53100 Siena, Italy; paolo.governa@unisi.it (P.G.); fabrizio.manetti@unisi.it (F.M.); 5Research & Development Agency of Aragon (ARAID) Foundation, 50018 Zaragoza, Spain; 6CIBER Enfermedades Respiratorias (CIBERES), Instituto de Salud Carlos III, 28029 Madrid, Spain

**Keywords:** avermectins, selamectin, tuberculosis, *Mycobacterium tuberculosis*, drug repurposing

## Abstract

Avermectins are macrocyclic lactones with anthelmintic activity. Recently, they were found to be effective against *Mycobacterium tuberculosis*, which accounts for one third of the worldwide deaths from antimicrobial resistance. However, their anti-mycobacterial mode of action remains to be elucidated. The activity of selamectin was determined against a panel of *M. tuberculosis* mutants. Two strains carrying mutations in DprE1, the decaprenylphosphoryl-β-D-ribose oxidase involved in the synthesis of mycobacterial arabinogalactan, were more susceptible to selamectin. Biochemical assays against the *Mycobacterium smegmatis* DprE1 protein confirmed this finding, and docking studies predicted a binding site in a loop that included Leu275. Sequence alignment revealed variants in this position among mycobacterial species, with the size and hydrophobicity of the residue correlating with their MIC values; *M. smegmatis* DprE1 variants carrying these point mutations validated the docking predictions. However, the correlation was not confirmed when *M. smegmatis* mutant strains were constructed and MIC phenotypic assays performed. Likewise, metabolic labeling of selamectin-treated *M. smegmatis* and *M. tuberculosis* cells with ^14^C-labeled acetate did not reveal the expected lipid profile associated with DprE1 inhibition. Together, our results confirm the in vitro interactions of selamectin and DprE1 but suggest that selamectin could be a multi-target anti-mycobacterial compound.

## 1. Introduction

Tuberculosis (TB) is the leading cause of death among bacterial infections. In 2019, an estimated 1.4 million people died from TB, and 10 million people developed the disease [1]. The situation is aggravated by the emergence of multidrug resistant (MDR) and extensively drug resistant (XDR) strains, which are responsible for over half a million cases every year. Up until recently, MDR/XDR-TB therapy could take up to 24 months requiring the use of second-line drugs with more severe and frequent side effects that complicate patients’ adherence to the treatment. The introduction of novel classes of anti-tubercular agents over the last decade has allowed for the development of new more effective regimes to treat MDR- and XDR-TB [2,3,4]. However, the overall success rate of drug-resistant TB therapies remains low, as only 50% of MDR-TB and one third of XDR-TB cases have positive outcomes [1,5]. Drug-resistant TB is still the first cause of death due to antimicrobial resistance [6], and the World Health Organization establishes *Mycobacterium tuberculosis*, the causative agent of TB, as one of the top priorities for the development of new antimicrobials [7].

Drug repurposing, i.e., finding new applications for clinically approved drugs, is an alternative to the traditionally time and resource consuming drug discovery process [8]. Following this approach, the anti-parasitic family of the avermectins was found to kill *M. tuberculosis* and other mycobacteria, a surprising finding since they were traditionally believed to be inactive against bacteria [9,10,11]. Avermectins are a family of 16-membered macrocyclic lactones with broad spectrum anthelmintic activity initially isolated from the soil-dwelling *Streptomyces avermitilis* [12]. In humans, ivermectin has been used for over three decades in the prevention and control of onchocerciasis and lymphatic filariasis [13], and it is now proposed as part of novel strategies for malaria vector control [14]. Other members of the family are also widely used to treat endo and ectoparasite infections in livestock and pets [15]. Selamectin is a veterinary avermectin used in cats and dogs with improved pharmacokinetic properties and lower toxicity over other avermectins [10]. Selamectin is the avermectin of choice for Border Collie dogs, a breed that is highly susceptible to avermectins because of the lack of a functional P-glycoprotein at the blood–brain barrier level. In addition, it shows better potency than other avermectins against mycobacteria [9,16]. The mode of action of avermectins against parasites is well-known: they bind to glutamate-gated chloride channels in nerve and muscle cells, increasing permeability to chloride ions and eventually causing death by paralysis [17]. However, their anti-mycobacterial mode of action remains to be elucidated.

DprE1 is a decaprenylphosphoryl-β-d-ribose (DPR) oxidase, which acts together with DprE2 to provide decaprenyl-phosphoryl-β-d-arabinose (DPA). The epimerization of DPR to DPA is achieved by the oxidation of DPR to the intermediate decaprenylphosphoryl-2-keto-β-d-erythropentofuranose (DPX) by DprE1 and the subsequent reduction of DPX to DPA by DprE2 [16]. DPA is the sole precursor for the synthesis of the arabinan moiety of the mycolyl-arabinogalactan-peptidoglycan (mAGP) complex, which is unique of mycobacteria and closely related species conferring intrinsic resistance to several antibiotics [18,19]. DprE1 is essential for *M. tuberculosis* growth [20], and a well-established antitubercular drug target, being inhibited by more than 15 different pharmacophores including the family of benzothiazinones, currently in Phase II clinical trials [5,21,22]. 

In this work, we identified DprE1 as a possible target of avermectins in mycobacteria and characterized this activity using selamectin as a model avermectin. We found that selamectin is able to inhibit the activity of the DprE1 enzyme, binding it through hydrophobic interactions that involve the residue at position 282 in *M. smegmatis* (corresponding to *M. tuberculosis* DprE1 residue 275). However, lack of phenotypic translation of these findings suggests a multi-target activity of selamectin against mycobacteria.

## 2. Results

### 2.1. Mycobacterium tuberculosis DprE1 Mutants Are More Susceptible to Selamectin

The Minimal Inhibitory Concentration (MIC) of selamectin was determined against a set of *M. tuberculosis* harboring different known mutations in genes encoding for drug targets (NTB1, DR1, 88.7), drug activators (53.3 and 81.10), or drug inactivator (Ty1) (Table 1). Resistance to selamectin was not observed; however, NTB1 and NTB9 strains displayed a consistent two-fold increase in sensitivity to the compound. These strains carry the mutations C387S and C387G in DprE1, respectively, conferring high-level resistance to benzothiazinones [22]. By contrast, the overexpression of *dprE1* gene (H37Rv-pSODIT-2:*dprE1* strain) did not confer changes in the susceptibility to selamectin (Table 1).

Time–kill kinetics assays were performed to confirm the increase in susceptibility of NTB1 *M. tuberculosis* mutant to selamectin, compared to the reference parental H37Rv strain. Selamectin was more active against NTB1 as illustrated by a marked CFU drop at earlier time-points. While the limit of detection (1-log CFU/mL) was reached at day 11 when treating the NTB1 strain, this was only reached at day 22 in the case of the H37Rv control strain. In addition, lower concentrations (8 µg/mL) of selamectin were more active against the NTB1 strain compared to the control strain (Figure 1).

### 2.2. Avermectins Inhibit the DprE1 Enzyme In Vitro

Two *M. tuberculosis* strains carrying mutations in DprE1 were more sensitive to selamectin (Table 1). To evaluate whether this enzyme could be involved in the mechanism of action of avermectins, the in vitro effects of the compounds were assessed against the DprE1 enzyme. Four avermectins were thus tested against the purified *Mycobacterium smegmatis* DprE1 recombinant enzyme. All of them were significantly active against DprE1, being selamectin the best inhibitor with an IC_50_ of 2.6 ± 1.1 µM (Table 2).

The two *M. tuberculosis* NTB1 (C387S) and NTB9 (C387G) mutant strains carry a mutation in the Cys387 residue of the DprE1 enzyme, known to be responsible for the binding to BTZ-043 [22]. To clarify the role of this residue in the inhibition by avermectins, the corresponding Cys394 in the *M. smegmatis* DprE1 enzyme was mutated to Ser (C394S), Gly (C394G) and Ala (C394A), the enzymes produced and purified. Mutation C394A was also investigated since all mycobacteria harboring this residue are resistant to BTZ [22]. 

The activity of the C394G mutant enzyme was negligible, as already reported [28], while C394S and C394A were active, but showed impaired kinetic parameters, leading to about 5-fold lowered specificity constants with respect to the wild-type (Table 3). Selamectin was able to bind both mutant enzymes, showing IC_50_ in the same range (C394A) or slightly higher (C394S) than the wild type enzyme. This suggests that, although avermectins inhibit the activity of DprE1 (Table 2), binding to the DprE1 active site is not primary related to the cysteine residue, similarly to other non-covalent inhibitors [29,30].

### 2.3. In Silico Model of Selamectin–DprE1 Binding

Evidence above described showed a secondary role of the *M. smegmatis* DprE1 Cys394 residue in the mechanism of inhibition of selamectin. Thus, to investigate alternative DprE1 binding sites of selamectin an in silico approach was used with a representative set of selamectin conformations docked into the structure of DprE1. Due to its high molecular size, selamectin was unable to fully penetrate the DprE1 binding site only occupying a portion of the ligand binding site close to the FAD cofactor, which accommodated the co-crystalized quinoxaline inhibitors of the X-ray crystallographic structures [30]. A large portion of the selamectin structure occupied a tunnel pointing toward the enzyme surface. An analysis of the best-scored binding poses showed that a hydrogen bond was formed between the oxime nitrogen atom and the guanidino moiety of the Arg325 side chain (Figure 2). The same amino acid is involved in a hydrogen bond with the nitro group of the quinoxaline inhibitor QN129 in complex with DprE1 (PDB 4P8T) (Appendix A). Moreover, part of both spiro and condensed heterocycles of selamectin were within the region occupied by the alkoxy-propanol ligand present in the complex [30]. Finally, the oxygen atom of the methoxy group of selamectin made an additional hydrogen bond with the hydroxyl substituent of the Tyr297 side chain, close to the solvent-exposed protein surface. It is worth noting that the lateral wall of the cavity included Leu275. 

Molecular dynamics simulations were also performed to evaluate the time-dependent stability of the selamectin–DprE1 complex. The evolution of the root-mean square deviation of 3D coordinates of selamectin atoms during MD simulation time in comparison to the last frame of minimization showed a 1.28 Å mean value with a 0.22 standard deviation, thus suggesting a significant stability of the complex. Moreover, a cluster analysis of the MD production phases aimed at evaluating the most frequent interactions of selamectin confirmed the hydrogen bonds with both Arg325 and Tyr297.

Even if the primary sequence of DprE1 belonging to different mycobacteria is conserved in the region that constitutes the binding site, sequence alignment revealed Leu275 polymorphisms in different species: *M. avium* carries a Val; *M. bovis*, *M. tuberculosis*, *M. smegmatis* and *M. abscessus*, a Leu; and *M. kansasii*, a Phe (Figure 3 and Appendix A). Increasing the bulkiness and lipophilicity of the amino acid chain from an isopropyl group of Val to an isobutyl chain of Leu to a benzyl moiety of Phe could enhance the interaction between selamectin and DprE1, therefore increasing the susceptibility of the enzyme to the ligand. In the last case, the aromatic side chain of Phe275 would be accommodated in a region of space that contains a cage of additional aromatic residues, such as Phe279, Phe320, and Trp323. The interactions between selamectin and DprE1 could be facilitated by the presence of Phe275 that enhances the hydrophobic contacts close to the binding site when compared to Val and Leu. Thus, such aromatic machinery could be responsible for the stabilization of the interaction between selamectin and DprE1.

### 2.4. Position Leu282 Is Involved in the Binding of Selamectin to DprE1 but Lacks Molecular to Phenotypic Translation

To verify our hypothesis about the binding of selamectin to DprE1, two *M. smegmatis* DprE1 mutant enzymes were constructed introducing a Val or a Phe at the position 282 (corresponding to *M. tuberculosis* DprE1 residue 275). The introduction of these mutations did not affect the catalytic activity of the enzyme since the kinetic parameters of the mutants were similar to those of the wild-type DprE1 (Table 3). By contrast, they displayed different responses to selamectin; the L282F mutant displayed an IC_50_ ca. 2-fold lower and the L282V mutant an IC_50_ ca. 20-fold higher than that of the wild-type (Table 3 and Figure 4). These data validated the *in silico* predictions and confirmed the importance of this residue at position 282 to stabilize the binding of selamectin to DprE1 through hydrophobic interactions. 

To ascertain whether the antimycobacterial activity of selamectin relies on DprE1 inhibition, the L282F and L282V mutations were then introduced in *M. smegmatis* DprE1 using single-stranded DNA recombineering to assess the molecular to phenotypic translation of selamectin activity (Appendix A).

MIC assays were performed against the mutant strains with no changes in their susceptibilities to selamectin compared to wild-type *M. smegmatis* (Table 4); these observations were also confirmed by time–kill kinetic assay (Appendix A). 

The effects of selamectin on the mycobacterial lipid composition were also investigated by metabolic labeling. Treatment of *M. tuberculosis* H37Rv or *M. smegmatis* mc^2^155 cells with selamectin at final concentrations of 4, 8, and 20 µg/mL or 4, 8, 16, and 40 µg/mL, respectively, did not result in significant changes in the lipid composition associated with the inhibition of arabinan synthesis. A general decrease in ^14^C uptake was observed upon selamectin treatment (Figure 5).

Altogether, these data suggest that, although the enzyme is efficiently inhibited by selamectin, DprE1 might not be the primary target and supports the hypothesis of avermectins being multi-targeting compounds.

## 3. Discussion

Drug repurposing is gaining traction across the pharmaceutical industry, given its potential advantages over the de novo drug discovery programs [9] in terms of time and economic investments. Drug repurposing is especially relevant in the case of underfunded diseases, such as tuberculosis, or neglected diseases where revenues are not expected to compensate the investment effort. Moreover, antimicrobials in general are also perceived as less profitable than other drugs because they are used to treat acute diseases and, since their lifespan is jeopardized by the emergence of drug resistant strains, new antibiotics are kept as last-resort options to preserve their activity. Indeed, repurposing efforts have already broadened our armamentarium to treat MDR- and XDR-TB, such as linezolid or carbapenems [2,5].

Linezolid and carbapenems target structures that are highly conserved amongst bacteria (the ribosome and penicillin-binding proteins) and thus, their mode of action can be assumed to be similar across bacteria. However, this is not the case for avermectins; although their mode of action has been defined for helminths [17] it is unlikely that it translates into mycobacteria. Thus, to get insight into the mechanism of action of selamectin, we screened a panel of *M. tuberculosis* strains with well-characterized mechanisms of resistance to a variety of avermectin molecules. We found that, while none of the mutations conferred resistance to selamectin, mutant strains NTB1 and NTB9 with C387S and C387G mutations in DprE1, respectively, were more susceptible (Table 1), suggesting a potential role for DprE1 in the mode of action of selamectin. 

DprE1 is a promising target for the development of new anti-tubercular drugs, with chemical scaffolds targeting this enzyme currently moving into the clinical studies [21]. Thus, we delved into the potential interaction between the avermectins and DprE1 by testing their activity against the purified enzyme. Our data confirmed that avermectins inhibit DprE1 in vitro, although with lower potency than benzothiazinones, being selamectin the most active with an IC_50_ of ca. 2.6 µM (Table 2). In addition, the mutant enzymes carrying the mutations present in the selamectin susceptible NTB1 (C387S) *M. tuberculosis* strain (which confer high-level resistance to BTZ043) and, especially, the mutation harboring an Ala residue in this position (C387A) showed selamectin IC_50_ values slightly different to that of the wild-type (Table 3). It is noteworthy that all the mutant enzymes displayed significantly altered catalytic properties. In the light of these results, we could hypothesize that the phenotypic effect observed in the NTB1 and NTB9 *M. tuberculosis* strains is the result of a reduced catalytic efficiency displayed by the mutant DprE1 forms, which would render *M. tuberculosis* more vulnerable to other chemical interactions with DprE1. Thus, lower concentrations of selamectin would suffice for activity below a critical threshold, as demonstrated by time–kill kinetic assays (Figure 1). 

Cysteine 387 is essential for the activity of covalent inhibitor of DprE1, such as benzothiazinones, but usually this residue is not the primary interaction with reversible inhibitors; this is the case of avermectins. Thus, to investigate other possible binding sites of selamectin, we performed in silico studies. Docking revealed a potential site in a cavity where known compounds block DprE1 through non-covalent inhibition [30]. Protein sequence in this region of the DprE1 enzyme was highly conserved among different mycobacteria (Appendix A) and shows almost no variation except for Leu275. Our finding that the moiety at the position 275 influences selamectin binding to DprE1 was consistent with the previous reports of selamectin activity against mycobacteria: *M. avium*, which carries a Val residue (with the smallest side chain in the series) at the 275 position is the less sensitive (MIC = 16 µg/mL); *M. bovis*, *M. tuberculosis*, and *M. smegmatis* carry a Leu and show a medium sensitivity to selamectin (MIC = 2–4 µg/mL); *M. kansasii*, bearing the largest amino acid (Phe) is the most sensitive to selamectin (MIC = 0.5 µg/mL) [9,11]. *M. abscessus*, carrying a Leu, represents an exception with an MIC > 32 µg/mL [11]; however, *M. abscessus* DprE1 carries an extra N-terminal 40 residues domain and shows the lowest sequence conservation (Appendix A).

To fully discern the actual role of the different amino acids at position 275 (282 in the *M. smegmatis* sequence) in the anti-mycobacterial activity of selamectin, we performed biochemical and genetic studies with *M. smegmatis.* Firstly, we produced the recombinant DprE1 mutant proteins in which the Leu residue at position 282 was replaced by Val and Phe. Both enzymes showed differences in their affinities to selamectin, demonstrating that this residue is involved in hydrophobic interactions with the compound. Indeed, the substitution of the Leu with the small Val led to a significantly lowered (20-fold) affinity to selamectin, indicating that this residue is unable to correctly interact with the compound. This was confirmed by the introduction of the Phe residue, which led to a 3-fold decrease in the IC_50_, confirming that a bulkier and more hydrophobic residue stabilizes the binding (Table 3).

Nevertheless, when the same mutations were introduced in the *M. smegmatis* genome, these did not translate into differences in the susceptibility to selamectin (Table 4). Moreover, despite the good activity of the compound against the in vitro enzyme activity, treatment with selamectin did not lead to alterations in the cell wall composition typical of DprE1 inhibition. Indeed, compounds inhibiting DprE1 activity in mycobacterial cells block the synthesis of the cell wall arabinan. The decrease of an arabinan acceptor for mycolic acids attachment leads to diminished amount of cell wall bound mycolates and accumulation of mycolic acids in the forms of trehalose monomycolates and trehalose dimycolates [31]. In our case, although a general decrease in ^14^C uptake was observed upon selamectin treatment, metabolic labeling of selamectin-treated *M. smegmatis* and *M. tuberculosis* did not show the changes in the lipid profile typically associated to the inhibition of DprE1 (Figure 5).

In this study we demonstrated that avermectins are inhibitors of the DprE1 enzyme, highlighting the role of the amino acid at positions 275 (in *M. tuberculosis*) in stabilizing their interaction at the molecular level. This is the first time that this residue is described to have a role in the inhibition of DprE1 activity. However, lack of molecular to phenotypic translation of DprE1 inhibition indicates that DprE1 could be only one among many factors underlying selamectin activity in mycobacteria. Avermectins are conceivably not covalent inhibitors of DprE1, and show overall relatively high IC_50_ values, compared to other DprE1 inhibitors that display activity in the nM concentration range [21]. Therefore, the potency of the interactions might be too low to translate the in vitro activity into differential growth inhibition, concealing the effect of inhibition under normal growth conditions. Indeed, *M. tuberculosis* NTB1 and NTB9 mutant strains (with a DprE1 enzyme highly compromised in their catalytic efficiency) are more sensitive to avermectins. In this case the intrinsically lower activity of the mutant enzymes could result inadequate to sustain bacterial viability in the presence of inhibitors at a certain concentration range (Figure 1).

As the avermectins have a clear phenotypic inhibitory activity against mycobacteria (with MIC values in the low µM range) these data strongly suggest alternative DprE1-independent molecular target(s). The multi-targeting nature of the avermectins might not come as a surprise considering not only their anthelmintic properties, but also antimicrobial, antiviral, antifungal, as well as anticancer and anti-diabetic activities [32]. In this context, multi-targeting or poly-pharmacology approaches to develop new anti-tubercular drugs are gaining interest among the drug developers. Indeed, the use of a single molecule acting on different targets has several benefits, such as possible lower cytotoxicity, lower rate of resistance development, and higher therapeutic efficacy, compared to the classical approach [33]. Together with the future identification of other mycobacterial targets, our finding paves the way for a rational design to improve their potency against DprE1 and develop a new generation of avermectins.

## 4. Materials and Methods

### 4.1. Bacterial Strains, Reagents, and Culture Media

*Mycobacterium smegmatis* and *Mycobacterium tuberculosis* were routinely grown in Middlebrook 7H9 broth (Difco, Sparks, MD, USA) supplemented with 10% Middlebrook Oleic acid-Albumin-Dextrose-Catalase (OADC) or ADC (Difco) and 0.05% (*v/v*) Tyloxapol or 0.025% Tween80 (*v/v*) or on Middlebrook 7H10 or 7H11 agar supplemented with 10% (*v/v*) OADC (Difco). For plasmid-carrying strains, hygromycin B (Invivogen, San Diego, CA, USA) was added at 20 µg/mL. Ivermectin, milbemycin oxime, moxidectin, and selamectin were purchased from the European Pharmacopoeia.

### 4.2. Drug Susceptibility Testing

Two-fold serial dilutions of the compounds were prepared in 7H9 broth supplemented with 0.2% glycerol and 10% ADC in 96-well flat-bottomed plates and seeded with *M. smegmatis* at a bacterial density of 10^5^ CFU/mL. Plates were incubated for 3 h before addition of 30 µL of the bacterial growth reporter MTT [3-(4,5-dimethylthiazol-2-yl)-2,5-diphenyltetrazolium bromide] (2.5 mg/mL) (Sigma, Madrid, Spain) and Tween 80 (10%, *v/v*) (Scharlau, Barcelona, Spain). The optical density at 580 nm was measured after further 3 h incubation, the lowest compound concentration that inhibited 90% MTT conversion to formazan was used to define the MIC value. 

*M. tuberculosis* strains were tested in duplicate in 96-well flat-bottomed plates in 7H9 with 10% OADC in a final volume of 200 µL. Bacteria were incubated for 7 days in the presence of the compounds before the addition of 10 µL of 0.025 mg/mL resazurin. Following 1-day incubation, compound activity was determined by fluorescence measures (530 nm excitation, 590 nm emission). 

### 4.3. Time–Kill Kinetics Assays

A starting inoculum of ca. 10^5^ CFU/mL of *M. tuberculosis* was pre-incubated in 7H9 with 10% OADC for 3 days at 37 °C before the addition of 0.1, 0.5, 2, 8, or 32 µg/mL of selamectin (corresponding to 0.025, 0.125, 0.5, 2, and 16-fold the MIC against *M. tuberculosis* H37Rv). Then, 10-fold serial dilutions were plated onto 7H11 agar with 10% OADC at 0, 1, 3, 7, 11, 14, and 22 days after compound addition, and CFUs enumerated after three weeks of incubation at 37 °C. 

For *M. smegmatis*, 10 mL of 7H9 with 0.2% glycerol (*v/v*) and 10% ADC were inoculated with *M. smegmatis* mc^2^155 cells from liquid cultures in late exponential phase to a final density of 10^5^ CFU/mL. Cultures were grown at 37 °C in the presence of 4 µg/mL and 16 µg/mL of selamectin (1 or 4-fold the compound MIC against the wild-type *M. smegmatis*) and 4 µg/mL and 16 ng/mL (0.5 or 2-fold the compound MIC against the wild-type *M. smegmatis*) and aliquots (100 µL) taken at 0, 24, 48, 72, and 96 h post inoculation. Then, 10-fold serial dilutions were seeded onto Luria–Bertani (LB) agar plates and CFUs enumerated after 3 days of incubation at 37 °C.

### 4.4. Docking of Selamectin to DprE1 and Molecular Dynamics Simulations

Twenty-nine three-dimensional structures of unbound or complexed DprE1 from *M. tuberculosis* or *M. smegmatis*, obtained by means of X-ray crystallographic experiments and characterized by a resolution ranging from 1.79 to 3 Å, are available within the Protein Data Bank. However, while the coordinates of 4P8L (sequence Gly5-Leu461, 2.02 Å resolution) and 4P8T (sequence Ala6-Leu461, 2.55 Å resolution) are completely solved [25], the remaining structures showed unsolved regions. Although with slightly lower resolution, 4P8T was used for computer simulations because it accommodated a non-covalent quinoxaline derivative (namely, QN129) bulkier than the quinoxaline Ty36c bound to DprE1 in the complex 4P8L [25]. The three-dimensional coordinates of the complex between DprE1 and the 3-[(4-cyanobenzyl)amino]-6-(trifluoromethyl)quinoxaline-2-carboxylic acid QN129 (PDB 4P8T) were submitted to the Protein Preparation Wizard of the Maestro suite [34] to assign bond orders, add hydrogens to the appropriate positions, and remove water molecules. Next, the Receptor Grid Generation routine was applied to the complex with the aim of codifying the physico-chemical properties of amino acids that constitute the DprE1 active site. Hydroxyl groups (in Tyr and Ser residues) within the grid were free to rotate. A Monte Carlo Multiple Minimum conformational search was applied to selamectin using OPLS3e as the force field and water as the solvent. Conformers were minimized with the Polak–Ribiere conjugate gradient until convergence on movement (0.05 Å threshold). Molecular docking simulations were performed by means of Glide software [35], with standard precision and flexible ligand sampling.

Following molecular docking calculations, molecular dynamics (MD) simulations were performed to check for the stability of the best-ranked docking complex using the Desmond software [36]. Before simulations, docked complex was solvated in a box of explicit water molecules (a TIP3P explicit solvent model) containing counterions to neutralize the charge. The overall system was then relaxed by energy minimization, heated to 300 K, and equilibrated to finally produce a 100 ns MD trajectory. Default settings were used for all other parameters.

### 4.5. DprE1 Enzymatic Studies

Recombinant *M. smegmatis* DprE1 was expressed in *E. coli* and purified as previously reported [21]. Mutants were generated using QuikChange site-directed mutagenesis kit, following manufacturer’s recommendations with the primers reported in Appendix A. Enzyme activity was determined with the Amplex Red/peroxidase coupled assay [28]. DprE1 was incubated in a reaction mixture containing 20 mM glycylglycine (pH 8.5), 500 µM farnesylphosphoryl-β-d-ribofuranose (FPR), 50 µM Amplex Red (Sigma-Aldrich), and 0.35 µM horseradish peroxidase. The formation of resorufin was followed at 560 nm (ε = 54,000 M^−1^ cm^−1^). Steady-state kinetic parameters were determined by assaying the enzyme at different FPR concentrations, and the kinetic constants determined by fitting the data to the Hill equation. Initial inhibition by all four avermectins was tested at 100 µM, using DMSO as negative control. IC_50_ values were subsequently determined according to the following equation:(1)$$A_{\left[I\right]}=A_{\left[0\right]}*\;\left(1-\frac{\left[I\right]}{\left[I\right]+IC_{50}}\right),$$
where *A*_[*I*]_ was the enzyme activity in the presence of a given concentration of inhibitor [*I*], and *A*_[0]_ the enzyme activity in the absence of inhibitors.

### 4.6. Construction of M. Smegmatis DprE1 Point Mutants

*M. smegmatis* mutants were constructed using single-stranded DNA recombineering as described previously [37]. Briefly, electrocompetent *M. smegmatis* pJV53H (a version of pJV53 [38] with kanamycin resistance replaced by hygromycin resistance) were electroporated with a mixture of 100 ng of a *rpsL* targeting oligonucleotide carrying a streptomycin-resistant mutation and 500 ng of the oligo carrying one of the mutations in *dprE1* (Appendix A). Following a four-hour recovery phase, bacteria were plated onto 7H10-OADC plates with 20 µg/mL streptomycin and incubated for 5 days. Colonies were screened for the presence of the point mutation by colony PCR followed by digestion of the PCR products with *Kpn*2I (ThermoFisher Scientific, Vilnius, Lithuania) for one hour. Due to the biological basis of recombineering, positive colonies are indeed a mixture of bacteria harboring the wild-type and the mutant *dprE1* alleles. Thus, colonies showing the undigested PCR product and the *Kpn*2I digestion products (Appendix A) were streaked onto agar plates, and colonies re-screened to eventually isolate the DprE1 mutants.

### 4.7. Analysis of Mycobacterial Lipids Using ^14^C Metabolic Labeling

*M. tuberculosis* H37Rv cells were grown in 7H9 broth supplemented with 10% albumin-dextrose-catalase and 0.025% Tyloxapol at 37 °C. Then, 100 μL culture aliquots (OD_600_ = 0.32) were added into Eppendorf tubes containing 2 μL of DMSO or 2 μL of selamectin stock solution. Final concentrations of selamectin in the cultures were 0, 4, 8, and 20 μg/mL. ^14^C acetate (specific activity 110 mCi/mmol, ARC) was added to each culture (final concentration, 0.5 μCi/mL) and cells cultivated for an additional 24 h.

*M. smegmatis* mc^2^155 cells were grown in 200 µL aliquots of 7H9 broth supplemented with 10% albumin-dextrose-catalase and 0.025% Tyloxapol at 37 °C. Selamectin was added during the culture inoculation at a final concentration of 0, 4, 8, 16, and 40 μg/mL. After 43 h of cultivation in 96-well plates, ^14^C acetate was added to each culture (final concentration, 0.2 μCi/mL) and cells were cultivated for an additional 6 h.

Whole cultures (100 µL or 200 µL) were transferred into 1.5 or 3 mL of chloroform/methanol (2:1) and lipids were extracted at 65 °C for 2–3 h. Samples were then subjected to biphasic Folch wash [39] (2×). Isolated lipids were dissolved in 30 µL of chloroform:methanol (2:1, *v/v*) and 5 µL loaded on thin-layer chromatography (TLC) silica gel plates F_254_ (Merck). Lipids were separated in chloroform/methanol/water (20:4:0.5, *v/v/v*) and visualized using an Amersham^TM^ Typhoon^TM^ Biomolecular Imager or by autoradiography.

## Figures and Tables

**Figure 1 ijms-23-00771-f001:**
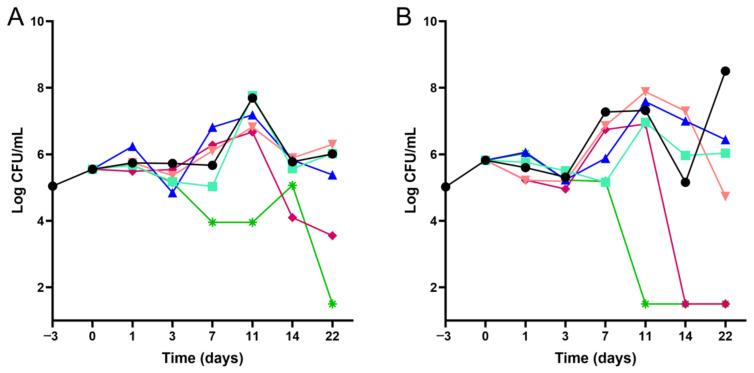
Time–kill kinetics of selamectin against wild-type *M. tuberculosis* H37Rv and NTB1 mutant strains. The 22-day kill kinetic assays were performed using various concentrations of selamectin (● 0 μg/mL; ■ 0.1 μg/mL; ▲ 0.5 μg/mL; ▼ 2 μg/mL; ♦ 8 μg/mL; ✴ 32 μg/mL) against the reference *M. tuberculosis* H37Rv (**panel A**) and NTB1 (**panel B**) strains.

**Figure 2 ijms-23-00771-f002:**
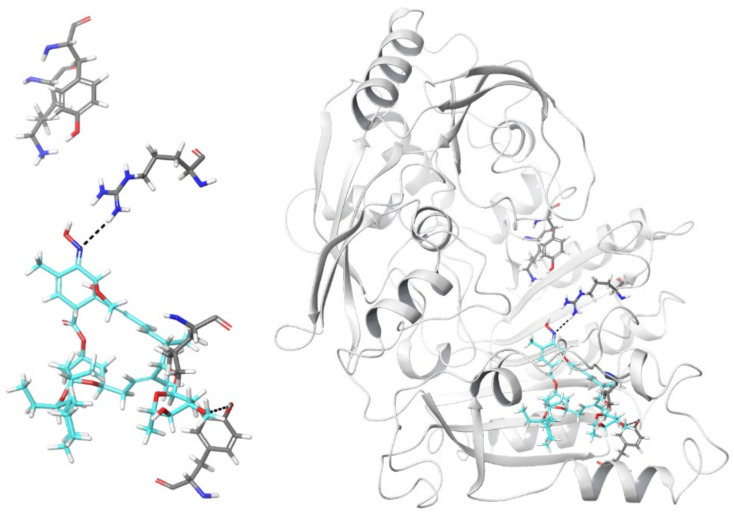
Graphical representation of one of the best-scored conformations of selamectin (thick tube representation, color by element + cyan carbons) within the DprE1 binding site (thick tube representation, **left panel**). For the sake of clarity, only a few amino acids are shown. Lys418, Tyr60, Arg325, Leu275, and Tyr297 (from left to right in the pictures) are represented by thick tubes. Hydrogen bonds that involve Arg325 (and the oxime nitrogen atom of selamectin) and Tyr297 (and the methoxy oxygen atom of selamectin) are shown by dashed black lines. The **right panel** shows the overall view of the selamectin/DprE1 docked complex.

**Figure 3 ijms-23-00771-f003:**
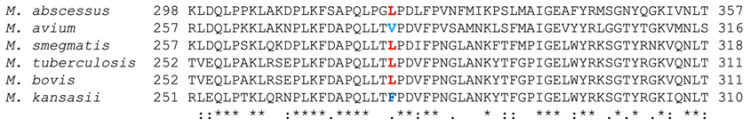
Multiple alignment of the DprE1 region around Leu275 (shown in red, variants in the same position are shown in blue). Asterisks indicate conserved amino acid residues in a given position in the protein sequence. Semicolons indicate moderate conserved position Full alignment is shown in Appendix A.

**Figure 4 ijms-23-00771-f004:**
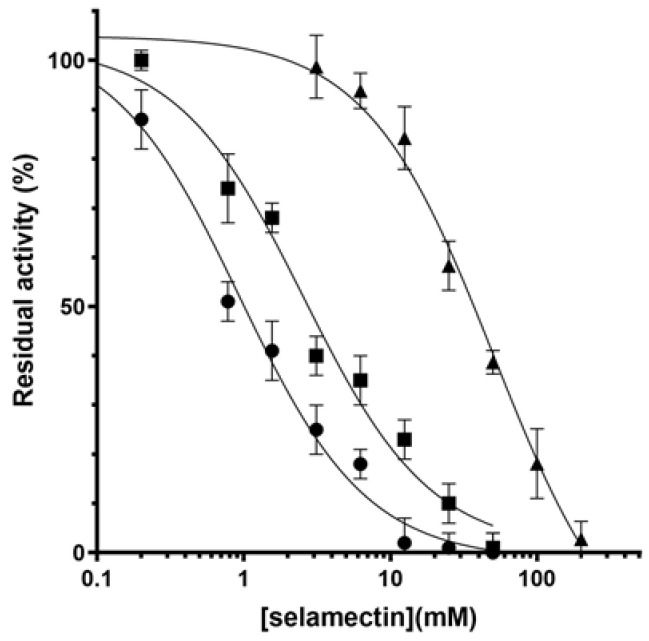
Dose response inhibition of DprE1 by selamectin. IC_50_ determination of selamectin against wild type (■), L282F (●), and L282V (▲) *M. smegmatis* DprE1.

**Figure 5 ijms-23-00771-f005:**
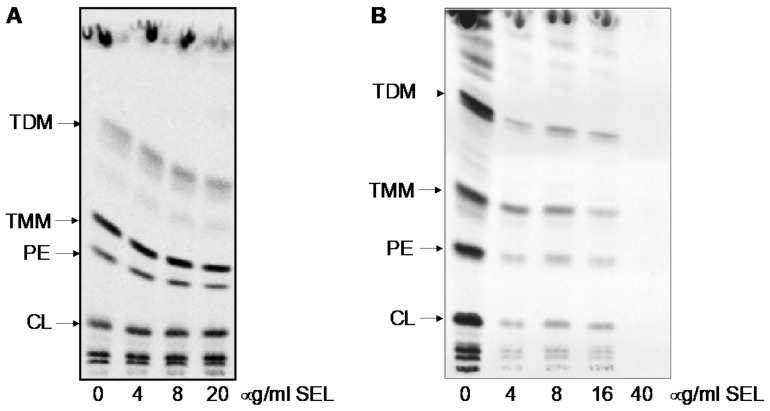
Analysis of mycobacterial lipid composition upon treatment with selamectin. Evaluation of the effect of selamectin (SEL) on the lipids of *M. tuberculosis* H37Rv (**A**) and *M. smegmatis* mc^2^155 (**B**) metabolically labeled with ^14^C acetate. Autoradiogram of TLC analysis of lipids separated in chloroform/methanol/water (20:4:0.5). TDM, trehalose dimycolates; TMM, trehalose monomycolates; PE, phosphatidylethanolamine; CL, cardiolipin; SEL, selamectin.

**Table 1 ijms-23-00771-t001:** MIC of BTZ043 and selamectin against a set of *M. tuberculosis* mutants resistant to a variety of anti-TB compounds. Isoniazid was included as internal control of activity. The “references” column indicates the first reports for the strains used in this study. INH, isoniazid; STR, streptomycin; RIF, rifampicin; EMB, ethambutol; PYR, pyridomycin; ETH, ethionamide; CAP, capreomycin; BTZ, benzothiazinones; nd: not determined.

M. tuberculosis Strain	Strain Resistance Profile	MIC INH (µg/mL)	MIC SEL (µg/mL)	MIC BTZ043 (ng/mL)	Reference
H37Rv		0.025	4	1	
H37Rv-pSODIT				1	
53.3 (Rv2466c, W28S)	TP053	0.025	2–4	nd	
53.8 (Rv0579, L240V)	TP053	0.025	2–4	nd	[23]
NTB1 (DprE1 C387S)	BTZ	0.025	2	>1000	[22]
NTB9 (DprE1 C387G	BTZ	0.025	2	1000	[22]
H37Rv-pSODIT:dprE1	BTZ	0.025	4	1000	
DR1 (mmpL3, V681I)	1,5-diarylpyrroles	0.025	2–4	nd	[24]
Ty1 (Rv3405c, c190t)	Ty38c	0.025	2–4	nd	[25]
88.1 (coaA, Q207R)	Thiophenecarboxamide derivatives	0.025	2–4	nd	[26]
88.7 (pyrG, V186G)	Thiophenecarboxamide derivatives	0.025	2–4	nd	[26]
81.10 (ethA, D1109-37)	Thiophenecarboxamide derivatives	0.025	4–8	nd	[26]
IC1	STR, INH, RIF, EMB	>0.2	4–8	nd	[27]
IC2	STR, INH, RIF, EMB, PYR, ETH, CAP	>0.2	4–8	nd	[27]

**Table 2 ijms-23-00771-t002:** Enzymatic activity of selected avermectins against the purified *M. smegmatis* DprE1. ^1^ BTZ data was obtained from reference [25].

Avermectin	IC_50_ (µM)
Ivermectin	13.2 ± 3.3
Milbemycin oxime	25.5 ± 5.6
Moxidectin	6.1 ± 0.9
Selamectin	2.6 ± 1.1
BTZ-043 ^1^	0.004

**Table 3 ijms-23-00771-t003:** Steady state kinetic parameters of *M. smegmatis* DprE1 and IC_50_ values of selamectin. SEL, selamectin; nd, not determined.

DprE1 Variant	*K_m_* (µM)	*k_cat_* (min^−1^)	*k_cat_/K_m_* (min^−1^ µM^−1^)	IC_50_ SEL (µM)
Wild-type	0.19 ± 0.20	5.1 ± 0.4	26.8 ± 1.1	2.6 ± 1.1
C394A	0.33 ± 0.22	2.8 ± 0.5	8.5 ± 0.6	3.9 ± 1.0
C394G	nd	nd	nd	nd
C394S	0.31 ± 0.15	1.6 ± 0.3	5.2 ± 0.6	12.1 ± 1.2
L282F	0.16 ± 0.12	4.5 ± 0.3	28.1 ± 1.4	1.0 ± 0.7
L282V	0.21 ± 0.15	4.3 ± 0.4	20.4 ± 1.1	50.6 ± 9.2

**Table 4 ijms-23-00771-t004:** MIC of BTZ-043 and selamectin against *M. smegmatis* point mutants at the Leu282 residue. SEL, selamectin.

M. *smegmatis* Strain	MIC BTZ-043 (ng/mL)	MIC SEL (µg/mL)
mc^2^155	8	4
mc^2^155-pJV53H	8	4
mc^2^155-pJV53H DprE1 L282F	8	4
mc^2^155-pJV53H DprE1 L282V	4	4

## Data Availability

All datasets generated for this study are included in the article/Appendix A.

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
