# Peer review of "The Veterinary Anti-Parasitic Selamectin Is a Novel Inhibitor of the Mycobacterium tuberculosis DprE1 Enzyme"

_ijms, 2022, doi:10.3390/ijms23020771_

Round 1

Reviewer 1 Report

.

Author Response

Answer to Reviewer 1 (IJMS-1477253)

This is a resubmission from an original one: IJMS-1424694.

In the original submission, three reviewers made comments. Two of them were positive about our work and, even thought an extensive point-by-point rebuttal letter to Reviewer 3’s comments was submitted explaining in depth and robustly confronting his/her arguments, the paper was rejected and we were asked by the Editor to resubmit, which we did (IJMS-1477253).

In the current version (resubmission IJMS-1477253), we have had another positive endorsement of our work by a forth reviewer (in this case Reviewer 2) who had no comments and even positively endorsed our work with sentences such as: "meticulous work to show in vitro inhibitory activity of this compound against DprE1" or "I appreciate authors giving a realistic conclusion that matches the results", which directly confronts Reviewer 3’s arguments.

In addition, in the current version (resubmission IJMS-1477253), we have had another reviewer (Reviewer 1) who is the same one as the Reviewer 3 from the original submission. It is unfortunate that in this new submission he/she declined to make any comments or answer our extended point-by-point rebuttal letter (which we are again copying and pasting at the end of this section) and just simply rejected it without making any comment supporting his/her decision. This attitude directly confronts the principles of the peer-reviewed system and raises questions about hidden intentions of this reviewer or even potential undisclosed conflicts of interest, which would discredit his/her opinion.

Without any further comments from Reviewer 1 (reviewer 3 in the original submission), we cannot modify our paper and we consider our second submission (IJMS-1477253) as valid and final.

At this point of the reviewing process, we request the Editor to make now a final decision based on the mounting evidence gathered so far. In summary, we have had four reviewers, three of them positively endorsed our paper and the fourth one, even though we provided extensive and robust point-by-point responses to his/her comments, has refused to provide any further rationale to support his rejection.

Sincerely

Santiago Ramón-García

P.S. Please, see below again the extensive point-by-point rebuttal letter previosly submitted to Reviewer 3 (IJMS-1424694) and that he/she has refused to reply.

Answer to Reviewer 3 (IJMS-1424694; Round 2)

"The author's comment above "since the Cys387 residue was not directly involved in the binding of the compound, we further investigate through the in silico studies the binding site of selamectin to DprE1, which was confirmed by mutagenesis studies." reflects the methodology of this paper. It has a line of thought but it is not consistent.

We would like to thank both the Reviewer and the Editor to give us the possibility to improve our manuscript. For this reason, we reorganized paragraph 2.2 that was consistently modified in order to better clarify the results that led to our conclusions. Moreover, we have modified the discussion to better explain our conclusions. Besides this, below we summarized our findings to provide a more clear view of the logical sequential steps carried out during our investigations:

  1. Avermectins are active against Mtb (Ref. 9-11)
  2. Mtb strains with mutation in the dprE1 gene are more sensitive to selamectin (Table 1 and Figure 1).
  3. Avermectins inhibit DprE1 enzymatic activity (Table 2)
  4. Binding of selamectin to the DprE1 active site is not primary related to the cystein residue (similarly to other non-covalent inhibitors) (Table 3)
  5. Selamectin binding to DprE1 involves the Leu275 residues (Fig. 2), which are stabilised through hydrophobic interactions (Figure 3).
  6. This binding is confirmed by dose-response enzymatic assays (Figure 4). The introduction of these mutations did not affect the catalytic activity of the enzyme since the kinetic parameters of the mutants were similar to those of the wild-type DprE1 (Table 3). By contrast, they displayed different responses to selamectin; the L282F mutant displayed an IC50 ca. 2-fold lower and the L282V mutant an IC50 ca. 20-fold higher than that of the wild-type, confirming that a bulkier and more hydrophobic residue stabilizes the binding,
  7. DprE1 cannot be the unique selamectin target since smegmatis L282 (corresponding to Mtb Leu275) mutants are not resistant to selamectin (Table 4), as well as the lipid biosynthesis was not affected (Fig. 5).

In our long experience in the study of the mechanism of action of antitubercular drugs we have often noticed how, for example, the over-expression of the target, as well as some target mutations do not confer drug resistance such as in the case of mmpL3 and for other multi-targeting compounds (e.g. PyrG/PanK inhibitors).

In conclusion, selamectin could be a multitargeting compound active in vitro against Mtb, being DprE1 one of its targets although not primary related to the Cys residue at the catalytic site. Interestingly, we identified Leu282 as the residue responsible for selamectin non-covalent binding; it is the first time that this residue is described to have a role in the inhibition of DprE1 activity.

Updated answers to Reviewer 3 (IJMS-1424694; Round 1)

Taking into account the Editor request, we would like to more carefully reply again to concerns raised by the Reviewer 3 in the first round of revisions. In particular, returning to the main points previously raised by the reviewer 3:

"Mycobacterium tuberculosis DprE1 mutants are more susceptible to selamectin" and "NTB1 and NTB9 strains 89 displayed a consistent two-fold increase in sensitivity to the compound." This is true but lacks relevancy. If selamectin was targeted to DprE, we would expect to see an increase in the MIC of NTB1 and NTB9 (for the site where BTZ binds).

As mentioned in our previous reply, a mutation in a drug target not necessarily led to an increase in MIC; depending on the effects of the mutation on the target and on its affinity for the compound these effects could be different. It is noteworthy that NTB1 and NTB9 mutant strains were not selected against avermectins, thus, there are no reason for which they should be resistant to them. However, the fact that they are more susceptible to the compounds could be ascribed to the mutation in DprE1. To confirm this hypothesis, we assayed the activity of avermectins against the purified DprE1 enzyme and its inhibition by the avermectin was confirmed by enzymatic assays.

We have modified accordingly the first part of paragraph 2.2.

This gets confusing when the authors say the selamectin's activity is "Cys387 independent"? So first it is told that selamectin is more active when C387 is mutated, but after its activity is C387 independent?

We understand that this paragraph could have created some confusion in the Reviewer 3’s interpretation of our results. Indeed, the first hypothesis was that these mutations led to an increased sensitivity of DprE1 to selamectin. We thus produced and studied the corresponding recombinant mutant enzymes. The mutants showed a slight difference in sensitivity to selamectin with respect to the WT enzyme. However, they showed a catalytic efficiency highly impaired, that could account for the lowered MIC of the mutant Mtb strains. In the discussion, we added our possible explanation for this. The fact that the mutant enzymes have similar levels of inhibition to the WT is not surprising. It is well-described that non-covalent DprE1 inhibitors do not primary involve the C394 in their binding mechanism. We modified the second part of paragraph 2.2 and the discussion accordingly.

Line 224 - "were more susceptible". This is true but lacks relevancy. I cannot agree with the authors when an assumption for target discovery is made upon a 2-fold decrease on an MIC of a mutated strain. This is exemplified in the unsuccessful results seen in lines 110 and 127.

As we explained above, this was our starting hypothesis, which led us to proceed with further experiment to verify that avermectins inhibit DprE1. Enzymatic assays indeed confirmed that the DprE1 activity is inhibited by avermectins.

"we performed in silico studies to investigate other possible binding sites of selamectin." So the authors assume DprE1 is the target of selamectin because of the reduced MIC in the mutants NTB1 and NTB9 (although, as explained before, I cannot agree with this).

As reported above, the assumption that DprE1 could be a target for the avermectins is not derived only from the reduced MIC of the mutants, but it is the result of sequential consecutive experiments with a robust logic behind them (described above); these experiments included MIC, TKA, enzymatic, docking and mutagenesis assays.

Then, it is confirmed that those mutations are not relevant for the activity of selamectin.

As mentioned in our previous rebuttal, a mutation in a drug target does not necessarily led to an increase in MIC. Indeed, depending on the effects of the mutation on the target and on its affinity for the compound these effects could be different. It is noteworthy that NTB1 and NTB9 mutant strains have not been selected against avermectins, so there is no reason for which they should be resistant. However, the fact that they are more susceptible to the compounds could be ascribed to the mutation in DprE1 and deficiency in their catalytic activity. To confirm this hypothesis, we assayed the activity of avermectins against the purified enzyme and confirm its inhibition by enzymatic assays.

After, the authors abandon completely their results/research so far to do a docking study on selamectin against the whole DprE1, and so binding energy results are shown. There are significant flaws in this study.

In silico experiment were not performed because “abandon completely our results/research so far” but, on the contrary, they follow a logic consecution of events. We first found that avermectins inhibited DprE1. However, we then found that the Cys394 was not the primary residue involved. Then, the next step was to investigate alternative binding sites by docking experiments, followed by site directed mutagenesis. We identified for the first time the Leu275 residue as an important residue for the interaction with selamectin, further confirming the specificity of the action of the compound against DprE1. We describe this at the beginning of paragraph 2.3 and in discussion

In conclusion, I cannot fully agree with the title of the manuscript. While evidence of binding is there (in silico and in vitro), selamectin is not a novel inhibitor of DprE1 as the activity of DprE1 was not hindered in any way.

Contrary to Reviewer 3’s opinion, we have clearly demonstrated by enzymatic assays that the activity of DprE1 is hindered by avermectins (Table 2) and that certain residues play a role in this binding (Table 3). The lack of in vitro (enzymatic) to in vivo (phenotypic) translation of results indicates a multi-target mode of action of the avermectins and that it cannot be assume that DprE1 is the only target of avermectins in mycobacteria. We provide some possible explanation at this regard in the discussion.

Reviewer 2 Report

Article entitled “The veterinary anti-parasitic selamectin is a novel inhibitor of the Mycobacterium tuberculosis DprE1 enzyme” is a meticulous work to show in vitro inhibitory activity of this compound against DprE1. It’s a good piece of work and presented very well. I also appreciate the authors giving a realistic conclusion that matches the results. I have no suggestions and thus recommend the article for publications after uploading the clean copy of the article.

Author Response

We appreciate Reviewer 2's comments and support to our manuscript, especially the comments about our "meticulous work to show in vitro inhibitory activity of this compound against DprE1" and how it is pointed out that "authors giving a realistic conclusion that matches the results", which was actually our intent to disseminate clear and realistic conclusions of our work. 

This manuscript is a resubmission of an earlier submission. The following is a list of the peer review reports and author responses from that submission.

Round 1

Reviewer 1 Report

The study aims to investigate the mode of action of avermectins, in particular selametacin, on Mycobacteria with the intention of explaining its tuberculicidal effect previously reported by some of the authors of this work. The study is part of an ongoing line of research of the group with international collaborators. The research is part of the drug repurposing trend applicable to an important pathogen. The work starts from a well-reasoned hypothesis, however the extensive efforts made with a brilliant and experimentally complex methodology show that selamectin could be a multi-target anti-mycobacterial compound not selective for the DprE1 enzyme as hypothesized.
It is a work that continues previous research of this group. The methodology is adequate and sufficient for the proposed objectives, the graphical representation is good and the conclusions of the discussion are acceptable. 
The manuscript is written with great clarity and can be easily understood by people who are not familiar with this field of research.

Author Response

We appreciate Reviewer 1’s comments and support to the conclusions of our work. The work to identify additional multi-target components is on-going and it will be the basis of future reports.

Reviewer 2 Report

Dear Authors, I have only minor corrections to the text. My main concern with the manuscript is the fact that the authors did not find the mechanism of action of selamectin. The paper revolves around a target and in the end, it was not the "correct" one. The authors could not perform additional experiments to get close to the target?

Minor corrections/suggestions:

Line 44 - Reference to the affirmation that DR-TB is the first cause of death due to AMR?

Line 63 - Reference to the "mode of action of avermectins" is missing.

Table 1 - If the results presented in Table 1 were obtained by the authors, why there is a references column, and what information is to be taken from these references?

Line 208 - Designating Tuberculosis as a neglected and underfunded disease is not correct. Fighting tuberculosis has been for decades one of the major investments by governments, who, etc. Maybe it's true for some countries, but not globally. 

Line 278 - Mycobacteria is not a genus or species name, it should not be in italics. 

Lines 328-331 - There is some missing information in this paragraph.

Author Response

Answer to Reviewer 2

We understand concerns raised by Reviewer 2. Currently we are dedicating several lines of research to elucidate the mode of action of selamectin. The one reported in here is one of these lines and future findings will be published in subsequent reports. Nowadays, it is becoming more and more accepted that antibiotics might have pleiotropic effects beyond a specific drug target. Here we demonstrated the binding of the avermectins to the DprE1 enzyme although this was not translated into phenotypic activity, probably due to the same pleiotropic/multi-target nature of the avermectins. We hope that when more evidence are gathered through our (and others) research, all together will give a global picture of how avermectins work against mycobacteria.

Minor corrections/suggestions:

Line 44 - Reference to the affirmation that DR-TB is the first cause of death due to AMR?

Answer: we have now included a reference. Now Ref. 6.

Line 63 - Reference to the "mode of action of avermectins" is missing.

Answer: we have now included a reference. Now Ref. 17.

Table 1 - If the results presented in Table 1 were obtained by the authors, why there is a references column, and what information is to be taken from these references?

Answer: data presented in Table 1 was generated by the authors. References relate to the original papers in which those strains were first reported. We have added text in the legend of Table 1 to clarify this point.

Line 208 - Designating Tuberculosis as a neglected and underfunded disease is not correct. Fighting tuberculosis has been for decades one of the major investments by governments, who, etc. Maybe it's true for some countries, but not globally. 

Answer: although it is true that much investment has been dedicated to TB research for decades, it is also true that it is clearly underfunded to reach the End TB WHO Strategy. With the COVID pandemic this has been strongly aggravated. In order not to lead to confusion we have re-structured the sentences and it now reads as follows:

Drug repurposing is especially relevant in the case of underfunded diseases, such as tuberculosis, or neglected diseases where revenues are not expected to compensate the investment effort.”

Line 278 - Mycobacteria is not a genus or species name, it should not be in italics. 

Answer: it has now been modified accordingly.

Lines 328-331 - There is some missing information in this paragraph.

Answer: we have now added the units missing.

Reviewer 3 Report

This paper by Ezquerra-Aznarez et al. suggests selamectin as a novel inhibitor of DprE1. I would suggest the authors to focus on some points described below:

Line 52 - There are a few papers that reported antimicrobial activity of avermectins (e.g. https://aricjournal.biomedcentral.com/articles/10.1186/s13756-018-0314-4). A brief mention to this and other papers may be needed to fully introduce the paper.

Line 85 - "Mycobacterium tuberculosis DprE1 mutants are more susceptible to selamectin" and "NTB1 and NTB9 strains 89 displayed a consistent two-fold increase in sensitivity to the compound." This is true but lacks relevancy. If selamectin was targeted to DprE, we would expect to see an increase in the MIC of NTB1 and NTB9 (for the site where BTZ binds). This gets confusing when the authors say the selamectin's activity is "Cys387 independent"? So first it is told that selamectin is more active when C387 is mutated, but after its activity is C387 independent?

Figure 1 - Quite confusing. Authors to use colours, other type of symbols to distinguish accurately the different concentrations.

Table 3 - An explanation on what L282F and L282V are would be important. There is no reference in the text above the table, why are they there?

Figure 2 - The overall form of the ligand and protein chosen is not the best. The reader cannot understand where is the protein selamectin is binding to.

Line 224 - "were more susceptible". This is true but lacks relevancy. I cannot agree with the authors when an assumption for target discovery is made upon a 2-fold decrease on an MIC of a mutated strain. This is exemplified in the unsuccessful results seen in lines 110 and 127.

Line 230 - No IC50 of BZTs reported in the table.

Line 144 - "we performed in silico studies to investigate other possible binding sites of selamectin." So the authors assume DprE1 is the target of selamectin because of the reduced MIC in the mutants NTB1 and NTB9 (although, as explained before, I cannot agree with this). Then, it is confirmed that those mutations are not relevant for the activity of selamectin. After, the authors abandon completely their results/research so far to do a docking study on selamectin against the whole DprE1, and so binding energy results are shown. There are significant flaws in this study.

Line 246 - What is the binding energy of DprE1 and selamectin. And that compared to the compound of the reference 22? Any similarities?

Line 256 to 257 - This conclusion may not be correct. We cannot assume that this difference "could explain these results". 

In conclusion, I cannot fully agree with the title of the manuscript. While evidence of binding is there (in silico and in vitro), selamectin is not a novel inhibitor of DprE1 as the activity of DprE1 was not hindered in any way.

Author Response

This paper by Ezquerra-Aznarez et al. suggests selamectin as a novel inhibitor of DprE1. I would suggest the authors to focus on some points described below:

Line 52 - There are a few papers that reported antimicrobial activity of avermectins (e.g. https://aricjournal.biomedcentral.com/articles/10.1186/s13756-018-0314-4). A brief mention to this and other papers may be needed to fully introduce the paper.

Answer: we appreciate the interesting suggestion coming from Reviewer 3 regarding the activity of ivermectin as antimicrobial. After careful review of the suggested manuscript we prefer to be cautious and not include such reference since only in 2 out of 22 isolates the inhibitory activity was observed and, in addition, time-kill assays showed no killing activity of the compound against those two isolates.

In comparison, the anti-mycobacterial killing activity of the avermectins has been demonstrated against a large panels of M. tuberculosis clinical isolates (Lim et al. 2013) and other mycobacterial species (Scherr et al. 2015 and Muñoz-Muñoz et al. 2021).

Line 85 - "Mycobacterium tuberculosis DprE1 mutants are more susceptible to selamectin" and "NTB1 and NTB9 strains 89 displayed a consistent two-fold increase in sensitivity to the compound." This is true but lacks relevancy. If selamectin was targeted to DprE, we would expect to see an increase in the MIC of NTB1 and NTB9 (for the site where BTZ binds). This gets confusing when the authors say the selamectin's activity is "Cys387 independent"? So first it is told that selamectin is more active when C387 is mutated, but after its activity is C387 independent?

Answer: a mutation in a drug target can lead to an increase in MIC values but depending on the effects of the mutation on the target and on its affinity for the compound, these effects could be different.

To investigate the MoA of selamectin we performed a screening of different Mtb strains carrying known mutations in genes coding already known targets. The fact that the MIC of the NTB1 and NTB9 was decreased indicates that DprE1 is involved to some extent in selamectin activity.

In our case, taking advantage of these findings, we demonstrated that avermectins inhibited the activity of the purified DprE1 enzyme. Thereafter, we assayed the avermectins against the DprE1 enzyme carrying mutations in the Cys387 residue to understand why this mutation was associated with increased sensitivity. However, the mutant enzymes did not show IC50 significantly different from the wild type. This result was not so surprising, and not in contrast with the MIC data. Indeed, the mutant enzymes showed catalytic efficiency significantly lower compared to that of the wild type, which implies that the DprE1 intracellular activity is conceivably lower in the mutant strains. Thus, a lower concentration of the compound is sufficient to bring DprE1 activity below the necessary threshold for cell survival. This was independent of the fact that the affinity (IC50 of the purified enzyme) of the wild type and the mutants for the compound were the same.

Finally, the fact that the inhibition is not directly related to the cysteine 387 is could be expected since this phenomenon is quite common for non-covalent DprE1 inhibitors. This last point has been better highlighted now in the text, including new references (Ref. 24 and 25) and now it reads as follows:

This suggests that binding of selamectin to the DprE1 enzyme, similarly to other non-covalent inhibitors is not directly related to this cysteine residue”.

Figure 1 - Quite confusing. Authors to use colours, other type of symbols to distinguish accurately the different concentrations.

Answer: we appreciate the comment for improvement and it has now been modified to a color-code version.

Table 3 - An explanation on what L282F and L282V are would be important. There is no reference in the text above the table, why are they there?

Answer: these data pertains to section 2.4. and references to it are clearly displayed:

“By contrast, they displayed different responses to selamectin; the L282F mutant dis- 175 played an IC50 ca. 2-fold lower and the L282V mutant an IC50 ca. 20-fold higher than that 176 of the wild-type (Table 3 and Figure 4).

Figure 2 - The overall form of the ligand and protein chosen is not the best. The reader cannot understand where is the protein selamectin is binding to.

Answer: Figure 2 has been updated in a simplified graphical representation.

Line 224 - "were more susceptible". This is true but lacks relevancy. I cannot agree with the authors when an assumption for target discovery is made upon a 2-fold decrease on an MIC of a mutated strain. This is exemplified in the unsuccessful results seen in lines 110 and 127.

Answer: although minor MIC changes, these were consistent, reproducible and confirmed by time-kill assays (Figure 1), thus supporting its relevance. The fact that DprE1 is inhibited by the avermectins, independently of the direct involvement of Cys387, additionally supports this relevancy.

Line 230 - No IC50 of BZTs reported in the table.

Answer: we appreciate this useful comment and the information has now been added to the table.

Line 144 - "we performed in silico studies to investigate other possible binding sites of selamectin." So the authors assume DprE1 is the target of selamectin because of the reduced MIC in the mutants NTB1 and NTB9 (although, as explained before, I cannot agree with this). Then, it is confirmed that those mutations are not relevant for the activity of selamectin. After, the authors abandon completely their results/research so far to do a docking study on selamectin against the whole DprE1, and so binding energy results are shown. There are significant flaws in this study.

Answer: the assumption that selamectin targets DprE1 is clearly not derived only based on the behaviour of the mutant strains but supported by the enzymatic assays that unequivocally showed that DprE1 is inhibited by avermectins. Since the Cys387 residue was not directly involved in the binding of the compound, we further investigate through the in silico studies the binding site of selamectin to DprE1, which was confirmed by mutagenesis studies.

Line 246 - What is the binding energy of DprE1 and selamectin. And that compared to the compound of the reference 22? Any similarities?

Answer: a comparison of the binding energy of selamectin and QN129 would provide minimal information in the context of this work since ligands to be compared are characterized by different structures (in terms of shape, size, and chemical features) and, what is more, by a partially different binding site (i.e., selamectin was predicted to occupy only a portion of the QN129 binding site due to its large molecular size that did not allow selamectin to penetrate this pocket in depth). Moreover, reporting the binding energy of only selamectin, without further comparison, would add little value to the study. Nevertheless, main and supplementary texts have been updated and a figure (now Figure S1) has been added to describe similarities between the selamectin and QN129 binding pockets.

Line 256 to 257 - This conclusion may not be correct. We cannot assume that this difference "could explain these results".

Answer: we appreciate the comment and the last sentence has now been removed.

In conclusion, I cannot fully agree with the title of the manuscript. While evidence of binding is there (in silico and in vitro), selamectin is not a novel inhibitor of DprE1 as the activity of DprE1 was not hindered in any way.

Answer: we regret that Reviewer 3 cannot fully agree with our findings. However, the title clearly reflects our findings and does not over-sell them. Selamectin has never been described as a DprE1 inhibitor so, being ours the first report, it is a novel finding and, in addition, the title limits the extend of our findings to the DprE1 “enzyme”, which its inhibition by the avermectins was clearly demonstrated by the enzymatic assays. In fact, these assays showed a significant concentration-dependent reduction of the activity of the enzymes in the presence of selamectin and the other avermectins. Moreover, the behaviour of the two mutated enzymes (in which we modified the leucine residue supposed to be responsible of hydrophobic interactions with selamectin) confirmed our hypothesis demonstrating a direct interaction within the compound and the protein.

Round 2

Reviewer 3 Report

No comments to add.